# Identification of Two Novel CIL-102 Upregulations of ERP29 and FUMH to Inhibit the Migration and Invasiveness of Colorectal Cancer Cells by Using the Proteomic Approach

**DOI:** 10.3390/biom11091280

**Published:** 2021-08-27

**Authors:** Kung-Chuan Cheng, Hsing-Chun Kuo, Meng-Chiao Hsieh, Cheng-Yi Huang, Chih-Chuan Teng, Shui-Yi Tung, Chien-Heng Shen, Kam-Fai Lee, Ya-Ling Yang, Ko-Chao Lee

**Affiliations:** 1Division of Colorectal Surgery, Department of Surgery, Chang Gung Memorial Hospital, Kaohsiung Medical Center, Kaohsiung 83301, Taiwan; topguncheng@gmail.com; 2Division of Basic Medical Sciences, Department of Nursing, Chang Gung University of Science and Technology, Chiayi 61363, Taiwan; guscsi@gmail.com (H.-C.K.); ccteng@gw.cgust.edu.tw (C.-C.T.); 3Research Fellow, Chang Gung Memorial Hospital, Chiayi 61363, Taiwan; 4Research Center for Food and Cosmetic Safety, College of Human Ecology, Chang Gung University of Science and Technology, Taoyuan 33303, Taiwan; 5Chronic Diseases and Health Promotion Research Center, Chang Gung University of Science and Tecnology, Chiayi 61363, Taiwan; 6Division of Colon and Rectal Surgery, Department of Surgery, Chang Gung Memorial Hospital, Chiayi 61363, Taiwan; mr8872@gmail.com (M.-C.H.); scorpievo@yahoo.com.tw (C.-Y.H.); 7Department of Hepato-Gastroenterology, Chang Gung Memorial Hospital, Chiayi 61363, Taiwan; ma1898@yahoo.com (S.-Y.T.); gi2216@adm.cgmh.org.tw (C.-H.S.); 8Graduate Institute of Clinical Medical Sciences, College of Medicine, Chang Gung University, Chiayi 61363, Taiwan; 9Department of Pathology, Chang Gung Memorial Hospital, Chiayi 61363, Taiwan; lkf2002@cgmh.org.tw; 10Department of Anesthesiology, Kaohsiung Chang Gung Memorial Hospital and Chang Gung University College of Medicine, Kaohsiung 83301, Taiwan; yaling453@yahoo.com.tw; 11Division of Colorectal Surgery, Department of Surgery, Chang Gung Memorial Hospital, Kaohsiung Medical Center, Chang Gung University College of Medicine, Kaohsiung 83301, Taiwan

**Keywords:** CIL-102, ROS, JNK, p300/CBP, ERP29, FUMH

## Abstract

CIL-102 (1-[4-(furo[2,3-b]quinolin-4-ylamino) phenyl]ethanone) is a major active agent of Camptotheca acuminata’s alkaloid derivative, and its anti-tumorigenic activity, a valuable biological property of the agent, has been reported in many types of cancer. In this study, we researched the novel CIL-102-induced protein for either the induction of cell apoptosis or the inhibition of cell migration/invasiveness in colorectal cancer cells (CRC) and their molecular mechanism. Firstly, our data showed that CIL-102 treatment not only increased the cytotoxicity of cells and the production of Reactive Oxygen Species (ROS), but it also decreased cell migration and invasiveness in DLD-1 cells. In addition, many cellular death-related proteins (cleavage caspase 9, cleavage caspase 3, Bcl-2, and TNFR1 and TRAIL) and JNK MAPK/p300 pathways were increased in a time-dependent manner. Using the proteomic approach with a MALDI-TOF-TOF analysis, CIL-102-regulated differentially expressed proteins were identified, including eight downregulated and 11 upregulated proteins. Among them, upregulated Endoplasmic Reticulum resident Protein 29 (ERP29) and Fumarate Hydratase (FUMH) by CIL-102 were blocked by the inhibition of ROS production, JNK activity, and p300/CBP (CREB binding protein) signaling pathways. Importantly, the knockdown of ERP29 and FUMH expression by shRNA abolished the inhibition of cell migration and invasion by CIL-102 in DLD-1 cells. Together, our findings demonstrate that ERP29 and FUMH were upregulated by CIL102 via ROS production, JNK activity, and p300/CBP pathways, and that they were involved in the inhibition of the aggressive status of colorectal cancer cells.

## 1. Introduction

Colorectal cancer (CRC) is an aggressive, malignant disease with a poor prognosis [1]. A large body of evidence demonstrates the self-sufficiency of CRC cells by using growth signals to escape from apoptosis, along with the tendency for cancer invasion and metastasis to be the fourth leading cause of cancer-related deaths in the industrialized world [2]. In fact, cell apoptosis is an important physiological process of cell death that helps maintain our body homeostasis [3]. Under the intrinsic apoptotic stimulus, such as an abnormal ROS evaluation, the release of cytochrome c from the mitochondria sequentially increases the activity of caspase-9 and caspase-3 for the induction of cell apoptosis [4]. In addition, ROS functions as the second messenger that is sensitive to oxidative damage-mediated cell apoptosis, and it triggers the intrinsic or extrinsic apoptotic signaling pathways [5]. Therefore, with regard to cancer therapy, the induction of cancer cell apoptosis, or ROS, is a great therapeutic strategy for destroying cancer cells without excessive inflammation [6]. Due to the issue of intrinsic chemo-resistance, it is imperative and urgent to develop more effective drugs for CRC treatment rather than classical chemotherapy treatments [7].

Recently, phenolic phytochemicals and Camptothecin (CPT)—alkaloids originally isolated from the bark and stem of Camptotheca acuminata—were both capable of inhibiting cell proliferation [8,9] and inducing cell apoptosis in CRC. CPT interacts mechanically with the DNA to form a complex, and it reduces the synthesis of DNA, RNA, and protein [10]. A number of the synthesized furo[2,3-b] quinoline derivatives, such as CIL-102, have been recognized for their anti-cancer effects on many cancer types, including prostate and breast cancer, leukemia, and cervical carcinoma [11,12]. CIL-102 treatment not only inhibits the proliferation and invasiveness of cancer cells [13,14] but also induces cancer cell apoptosis [15]. Many intracellular signals, such as ERK1/2, Cdc25cSer216, p21, GADD45, and ROS production, participate in CIL-102-mediated anti-cancer action [15,16]. In gastric cancer, CIL-102 treatment has shown a strong anti-cancer effect. Furthermore, the treatment activates intracellular signals for the induction of cancer cell apoptosis. These signals were involved in the H3K4 trimethylation of TNFR1 and TRAIL proteins, including ROS derived and JNK/mTOR/p300 pathways in DLD-1 cells. CIL-102 treatment works well to induce cell apoptosis in other types of cancers, such as colorectal cancer; however, its mechanism related to the induced downstream protein, by means of CIL-102 treatment for its anti-invasiveness properties, remains unclear.

By using the proteomic approach with the MALDI-TOF-TOF analysis (2DE MS/MS), Endoplasmic Reticulum resident Protein 29 (ERP29) and Fumarate hydratase (FUMH) were identified as the novel upregulation proteins of CIL-102 treatment in DLD-1 cells. Furthermore, CIL-102 increases and produces ROS, JNK activity, and p300/CBP pathways. Both ERP29 and FUMH are required for the CIL-102-mediated inhibition of cell migration and the invasiveness of colorectal cancer cells. Taken together, this study demonstrates a novel anti-cancer mechanism for CIL-102 to inhibit cell migration and the invasiveness of colorectal cancer cells via the upregulation of ERP29 and FUMH.

## 2. Materials and Methods

### 2.1. Chemical Reagents, Antibodies, and Cell Culture

The chemical reagents, such as 1-[4-(Furo[2,3-b]quinolin-4-ylamino)phenyl] ethenone (CIL-102), 3-(4,5-dimethylthiazol-2-yl)-2,5-diphenyltetrazolium bromide (MTT), ROS scavenger (*N*-acetyl cysteine [NAC]), 2,7-dichlorodihydrofluorescein diacetate (H_2_DCFDA), dihydroethidium (DHE), ERK inhibitor (PD98059), c-Jun N-terminal kinase (JNK1/2) inhibitor (SP600125), p300/CBP inhibitor (C646), mTOR inhibitor (rapamycin), SDS, NP-40, sodium deoxycholate, and protease inhibitor cocktails, were obtained from Sigma (St. Louis, MO, USA). The antibodies were obtained from Santa Cruz Biotechnology (Santa Cruz, CA, USA), including anti-p300/CBP (sc-32244), anti-Bcl-2 (sc-7382), anti-Bcl-XL (sc-8392), anti-β-actin (sc-8432) antibodies (diluted 1:1000), and monoclonal secondary antibodies (sc-2357, diluted 1:5000). The purchased antibodies from Cell Signaling Technology (Beverly, MA, USA) were anti-cdk1, anti- ERK1/2Thr^202^Tyr^204^ (#9101), anti-JNK1/2 Thr^183^Tyr^185^ (#9251), anti-cleavage caspase-3 (#9661), and caspase-9 (#9505) antibodies (diluted 1:1000). The purchased antibodies from Millipore (Millipore, CA, USA) were anti-Acetyl-Histone H3 (Lys9/14) (12-360) antibodies (diluted 1:1000). The purchased antibodies from Sigma-Aldrich (Sigma, Saint Louis, MO, USA) were anti-p300/CBP (P2859) and anti-β-actin (A5441) antibodies (diluted 1:1000). All culture materials were from Gibco (Grand Island, NY, USA). Two human colon cancer cell line DLD-1 (BCRC Number: 60132) and the human colorectal carcinoma cell line HCT-116 (BCRC Number: 60349) were purchased from the Bioresources Collection and Research Center (BCRC) of the Food Industry Research and Development Institute (Hsinchu, Taiwan). The human DLD-1 and HCT-116 cells were grown in plastic tissue culture flasks or dishes or in microplates (Nunc, Naperville, Denmark) with the cell culture medium, including Dulbecco’s Modified Eagle Medium (DMEM), and supplemented with 10% fetal calf serum (Gibco), non-essential amino acids, 1 mM sodium pyruvate, and 1% antibiotics (100 units/mL of penicillin and 100 μg/mL of streptomycin), at 37 °C in a humidified atmosphere of 5% CO_2_ and 95% air [17].

### 2.2. Cell Viability and Reactive Oxygen Species Detection

The 4′,6-diamidino-2-phenylindole (DAPI) staining was used to check the morphological characteristics of the cells under fluorescence microscopy. After fixing with 4% paraformaldehyde for 30 min at room temperature, permeabilizing in 0.2% Triton X-100 in phosphate-buffered saline three times for 15 min, and then PBS washing, the cells were incubated with 1 μg/mL of DAPI for 30 min. Using a fluorescent microscope with 340/380 nm, the apoptotic nuclei in the field of the 200~300 cells were observed. According to a previous report [18], the percentage of apoptotic cells was scored and calculated under a 200× magnification excitation filter.

As previously described, after co-staining with Annexin V–FITC and propidium iodide (Biosource International, Camarillo, CA, USA), the cells were subjected to flow cytometer analysis (Attune NxT Flow Cytometer, Thermo Fisher Scientific Inc.) for measurement of cell apoptosis. Based on the fluorescent intensity, the number of the apoptotic cells (V+/PI−) can be calculated using the Attune NxT software 3.1 (Thermo Fisher Scientific Inc., Ramsey, MN, USA), and the data are represented as a percentage of the untreated control group with three independent experiments [19].

Using the fluorescent probes of H2DCFDA (2,7-dichlorodihydrofluorescein diacetate), it was possible to detect intracellular accumulation of ROS (O^2−^) in the cells. After PBS washing, the cells were applied to Attune NxT Flow Cytometer analysis and analyzed with Attune NxT software. The data were analyzed and presented as a percentage of the fluorescent intensity with three independent experiments. The apoptotic cells (V+/PI−) were identified using the fluorescence-activated cell sorter analysis in an Attune NxT Flow Cytometer [20].

### 2.3. Matrigel Invasion and Scratch Analysis

Based on a chamber with two medium-filled compartments, the Boyden chamber assays were used to measure tumor cell invasion. As described above, the cells (1 × 10^5^/mL cells (1 × 10^5^/mL)) in a serum-free medium were collected and added to an inner cup of the 48-well transwell chamber (Corning Life Sciences, Corning, NY, USA). The transwell chamber was coated with 50 µL of matrigel (BD Biosciences, Franklin Lakes, NJ, USA; 1:10 dilution in a serum-free medium). The outer cup had the medium that was supplemented with 10% serum or the indicated agent. After 24-h incubation at 37 °C in a humidified atmosphere with 5% CO_2_, the membrane containing the cells was fixed and stained with a modified Giemsa stain (Sigma, Saint Louis, MO, USA). Under a light microscope at a 200× magnification, the cells on the lower side of the membrane were counted and analyzed [21].

Scratch assays of the plating cells in a six-well culture dish were performed. After the cells attached and reached confluence, a 4-mm scratch was made through the culture dish. After washing twice with phosphate-buffered saline (PBS, pH = 7), the cells were cultured in the culture medium with or without CIL-102. Using Openlab v3.0.2 image analysis software (Improvision, Coventry, UK), it was possible to quantify the area that was progressively filled with the cells in the period of the experimental time [21].

### 2.4. Proteomic Dimensional Protein Electrophoresis Analysis

The chemical reagents and experimental procedure for 2D gel electrophoresis were described in our previous study [22]. After treatment, the total proteins of the cells were extracted and precipitated by 10% trichloroacetate in acetone. Their concentrations were measured by the Bradford assay with bovine serum albumin as the standard sample for normalization. Prior to the 2D-PAGE analysis, protein samples were suspended in a rehydration solution and then applied to Iso-Electric Focusing (IEF) in the pH 3–10 immobilized-gradient strips (Immobiline Dry Strips, Amersham Biosciences, Uppsala, Sweden) with an Ettan IPGphor II apparatus (Amersham Biosciences). Using 10% SDS-PAGE gels, it was possible to carry out the two-dimension electrophoresis.

### 2.5. In-Gel Digestion and the Peptide Fingerprints’ Identification with MALDI-TOF

The total cell proteins that were resolved in six pairs of silver-stained 2D SDS-PAGE gels were scanned using the ImageMaster 2D Platinum Software 6.0 (Amersham Biosciences). Therefore, the protein profiles of each pair of silver-stained gels were recorded and compared with other treated groups. Among all six pairs of 2D gels, the protein spots differentially expressed by at least three folds were subjected to the in-gel digestion for further mass spectrometric analysis of matrix-assisted laser resorption ionization-time-of-flight flight/time-of-flight (MALDI-TOF/TOF). After the gel pieces were then dehydrated, performed with trypsin digestion, the FlexAnalysis system (Bruker-Franzen Analytik, Bremen, Germany) was used to acquire mass spectra as the sum of the ion signals by the irradiation of the targets with a mean of 300 laser pulses. Peptide fingerprints (selected in the mass range of 700–4000 Daltons) were analyzed by the Mascot software (http://www.matrixscience.com, accessed on 5 February 2021). A Mascot score with *p* < 0.05 was considered statistically significant, as described in our previous study [22]. The MALDI-TOF/TOF data were searched and analyzed by the in-house MASCOT software (version 2.2.04) to identify proteins that required the detection of unique peptides and proteins with more than two spectral counts. Then, by using MASCOT search engines, the peptide mass data of each spot were submitted to the SwissProt 100425 human species bio-information stations for further analysis. With a higher MASCOT score in the bovine database than in the human database, the proteins were considered as serum contamination and removed [23].

### 2.6. Cell Extracts’ Preparation and Immunoblot Analysis

To obtain the total cell lysate, the cells were lysed with a buffer (1% NP-40, 0.5% sodium deoxycholate, 0.1% Sodium Dodecyl Sulfate (SDS)) and a protease inhibitor mixture (phenylmethylsulfonyl fluoride, aprotinin, and sodium orthovanadate). As previously described [24], the total cell lysates (50 μg of protein) were separated by SDS-polyacrylamide gel electrophoresis (PAGE) (12% running, 4% stacking). After transferring the protein in gel on the membrane, the designated antibodies and Western light chemiluminescent detection system (Bio-Rad, Hercules, CA, USA) were used to detect the level of the specific proteins.

### 2.7. The shRNA Lentivirus Transfection

The methods of the shRNA Lentivirus transfection for applying the knockdown of genetic functions were established (Santa Cruz Biotechnology) [14]. Following the recommended protocol, the cells were infected with the designated Control shRNA Lentiviral Particles, ERP29 and PROF1 siRNA, and shRNA Plasmids using the shRNA expression of lentiviral particles (Santa Cruz, CA, USA).

### 2.8. Statistical Analysis

Data were reported as the means ± standard deviation (means ± SD) of three independent experiments, and the groups were compared using the one-way Analysis of Variance (ANOVA) with Tukey’s Multiple Comparison Test by the SAS software statistical package “SigmaPlot” version 9.0 (SAS Institute Inc., Cary, NC, USA) [25]. Significant differences were established at *p* < 0.05.

## 3. Results

### 3.1. CIL-102 Reduces the Migration and Invasion of DLD-1 Cells

Our previous study demonstrated that there was an increase in the cellular levels of p21 and GADD45 by the 9-anilinofuroquinoline derivative, CIL-102, inhibiting DLD-1 proliferation and cell cycle distribution [15,16]. It remains unknown whether it can inhibit migration and invasion in the CRC. Its affecting proteins also remain unknown. Firstly, DAPI staining showed that treatment with CIL-102 at 1-µM concentrations induced the early apoptotic chromatin condensation in DLD-1 cells (Figure 1A), but previously not in normal epithelial HCoEpiC cells [13]. Using the scratch-wound assay to observe the continuous rapid movement of DLD-1 cells for 24 h (Figure 1B), our data revealed that cells that did not receive the CIL-102 treatment control showed at a high confluence (90–100%) of the monolayer region, which gradually migrated into the cell-free “scratch” region (Figure 1D). In contrast, treatment with CIL-102 at 1- and 2-µM concentrations for 24 h reduced the cell migration by 22% and 11%, compared to the treatment control group (Figure 1C). Furthermore, by using the Boyden chamber assay, we determined that CIL-102 at 1-, 2-, and 5-µM concentrations also inhibited the invasiveness of DLD-1 cells. The quantification of revealed observations also exhibited a significant anti-invasive effect compared to the control group and was respectively shown as 65%, 35%, and 16%, which supported the results obtained with the scratch-wound assay in a dose–response relationship (Figure 1E). By using flow cytometry analysis for annexin-V and PI, our results revealed that an increased percentage of annexin V-positive cells was found in all untreated and CIL-102-treated DLD-1 cells in a dose-dependent manner and was shown as 8%, 23%, and 28%. (Figure 2A). In addition, CIL-102 initially increased the intracellular ROS production in DLD-1 cells after 24 h, by 1.2- and 1.5-fold (Figure 2B), compared to the control.

### 3.2. CIL-102 Treatment Triggers Apoptotic Signals in DLD-1 Cells

To determine which cell death-related proteins were involved in the CIL-102-induced cell apoptosis of DLD-1 cells, the protein levels of the hallmarks representing apoptotic cell death [26], such as the cleavage of caspase 9, caspase 3, and Bcl-2, were measured by Western blot. Our data showed that the CIL-102 treatment increased the active form of caspase-3 and -9 in a time-dependent manner in DLD-1 cells (Figure 3). In contrast, anti-apoptotic Bcl-2 and Bcl-XL proteins were decreased in the cells with the CIL-102 treatment (Figure 3A). Furthermore, in comparison to the untreated group, a time-dependent increase was found in the phosphorylation of JNK1/2 Thr183/Tyr185 and the level of p300/CBP in the CIL-102-treated DLD-1 (Figure 3B). Accordingly, our data showed that CIL-102 may activate multiple-protein kinases’ pathways for the induction of intrinsic cell apoptosis.

### 3.3. Proteomic Profiling of CIL-102-Treated DLD-1 Cells

By using the proteomic approach with the MALDI-TOF-TOF analysis, we wanted to investigate the novel CIL-102-upregulated protein [27,28]. More than 800 protein spots from cell lysates of DLD-1 cells, with or without the CIL-102 treatment, were visualized in the silver-stained 2D-PAGE analysis (Figure 4). The images of protein expression profile gels (i.e., the six pairs from the control and the CIL-102-treated groups) were analyzed by using the ImageMaster software. Our image analysis showed that 19 proteins were identified as differentially expressed proteins, where their protein level had more than a three-fold change compared to the untreated and CIL-102-treated groups (Figure 4). Among them, 11 protein spots showed a greater than three-fold change in the CIL-102 treatment (Figure 5, Table 1), including Fumarate Hydratase (FUMH) and Endoplasmic Reticulum Resident Protein 29 (ERP29). Compared to the untreated group, eight spots of 19 differential display proteins were downregulated consistently by the CIL-102-treated group (Figure 5). After the peptide fingerprint identification of these spots by MALDI-TOF MS, the full names of 19 proteins were listed in Table 1. It presents the zoomed views of a representative gel region and displays several differentially expressed, oxidative stress-related proteins on the effect of CIL-102, which inhibited invasive action in the DLD-1. Spots 9 and 10 were subsequently identified by using a 2D proteomic analysis, including Fumarate Hydratase (FUMH) and Endoplasmic Reticulum Resident Protein 29 (ERP29), which may mediate the oxidative stress and the aggressive effects of CIL-102 in DLD-1 cells (Table 1). From previous studies, one protein ERP29 was shown to be a tumor suppressor and a chaperone protein, which involves the regulation of primary tumor development and the arrest of cell growth [27]. Another differential display protein, FUMH, also shows anti-tumor functions, including the inactivation of tumor metastasis, tumor aggressiveness, and EMT changes by epigenetic modification [28].

### 3.4. Upregulation of FUMH and ERP29 by CIL-102 via the Signaling Pathways of ROS, JNK, and Histone Acetylation to Inhibit Cell Migration and Invasion

Next, we determined if FUMH and ERP29 play a role in the reduction of cancer cell migration and invasion by CIL-102. As shown in Table 2, the downregulation of both FUMH and ERP29 by shRNA significantly decreased the CIL-102-inhibited migration and invasion in DLD-1 cells at 24 h, respectively (* *p* < 0.01). In contrast, the addition of Lenti shRNA ERP29 and shRNA FUMH alone significantly increased cancer migration and invasion (Table 2), which implies that they have a tumor-suppressive effect. These data indicated that CIL-102 treatment reduced the migration and invasion of DLD-1 cells through the upregulated FUMH and ERP29.

To determine which signaling pathways are involved in the upregulation of FUMH and ERP29 by CIL-102, chemical inhibitors were used to block ROS production, JNK activation, and histone acetylation in CIL-102-treated DLD-1 cells. As shown in Figure 6, ROS scavenger NAC, JNK inhibitor SP600125, and P300 inhibitor C646 almost blocked the CIL-102-induced levels of the phospho-JNK, P300 CBP, and histone H3K9K14ac (acetyl Lys9/Lys14). Furthermore, our data showed that the increase of FUMH and ERP29 by CIL-102 was blocked by these chemical inhibitors (Figure 6), which suggests that the CIL-102 treatment increased FUMH and ERP29 expression through the production of ROS and the activation of JNK and P300. Taken together, our results demonstrated a novel mechanism for CIL-102 to reduce the migration and invasion of CRC cancer via the upregulation of FUMH and ERP29.

The results and the data showed that FUMH and ERP29 expression in DLD-1 cells is essential for the implication of oxidative stress ROS and JNK/P300 CBP signaling; this is along with the association that CIL-102 inhibited cell invasion and tumor growth. These results are consistent with the proteomic results, which indicates that a proteomic differential display model FUMH and ERP29 are applicable when assessing CIL-102-inhibited DLD-1 cells.

## 4. Discussion

Colorectal cancer (CRC) is a sequential multistage process that includes tumor initiation, tumor promotion, and tumor metastasis [29]. Effective agents for CRC treatment have not yet been found. Therefore, bioactive safe compounds from foods have been considered a source for developing the chemo-preventive compounds for cancer chemoprevention and metastasis suppression. Many reports demonstrate that these chemo-preventive compounds can also inhibit cancer cell metastasis [4,30,31] via a variety of mechanisms, such as ROS production. CIL-102, a derivative of the 9-anilinofuroquinoline from the bark and stems of Camptotheca acuminate, is used as an antiseptic drug [8,9]. Anti-cancer and chemo-preventive properties of CIL-102 have recently been reported, including the inhibition of tumor cell proliferation [10], the induction of cell apoptosis, and cell cycle arrest [11,12]. However, the biochemical effect of CIL-102 that triggers its anti-carcinogenic properties in CRC cells and its regulatory mechanism remains unclear. In this study, we found that CIL-102 reduces the aggressive status (migration and invasiveness) of DLD-1 cells (Figure 1). Furthermore, our data indicate the essential role of ROS generation and JNK/p300 CBP pathways during the execution of apoptosis and anti-invasion by using naturally extracted CIL-102 (Figure 2 and Figure 3B, Table 2). This study elucidated the mechanism of the observed inhibition of the metastasis of CRC cells, suggesting that the inhibition of metastasis is related to the CIL-102.

Many studies demonstrate that cellular mechanisms contribute to the overall cancer-prevention effects of these dietary phytochemicals [31,32]. Our previous study demonstrated that CIL-102 causes mitotic arrest and the tumor-growth inhibition of human CRC cells via ROS generation [14,15,16]. In this study, our data showed how the method of CIL-102 treatment may be used to inhibit the tumor invasion of DLD-1 cells through ROS increment and the activation of the JNK/p300 CBP signaling pathway. With respect to the dosage and duration of inductive ROS production, phenolic phytochemicals activate signal transduction pathways, leading to either cell cycle arrest or apoptosis and invasiveness [4,13,16].

Our present study demonstrates that DLD-1 cells treated with CIL-102 show oxidative alterations in terms of signaling transduction events and ROS overloading that result in the therapeutic effects on cancer invasion (Figure 2, Table 2). To identify the novel CIL-102-regulating molecule and its anti-cancer mechanism, two-dimensional electrophoresis (2-DE)-based proteomic analysis was performed (Figure 4). The level of Endoplasmic Reticulum resident Protein 29 (ERP29) and Fumarate Hydratase (FUMH) were increased in CIL-102-treated DLD-1 cells (Figure 4 and Figure 5). A recent study demonstrated that ERP29, a molecular chaperone, acts as a tumor-suppressor protein and a novel regulator, which leads to cell growth arrest and cell transition from a proliferative to a quiescent state. Furthermore, it leads to the reprogramming of molecular portraits to suppress tumor growth. Thus, ERP29 needs to be further assessed as a potential effect of medical intervention for CRC therapy by CIL-102 for inhibiting the malignant behavior of the CRC [33,34,35].

In addition, FUMH, an enzyme participating in the Tricarboxylic Acid (TCA) cycle, catalyzes the reversible hydration of fumarate to generate malate [36]. We also previously found the effect of CIL-102 on neuroblastoma cells and identified FUMH as having tumor suppressor- and tumor invasiveness-related proteins. CIL-102 elicits these proteins during the upregulation of FUMH. They play a central role in controlling the migratory potential of tumor cells by regulating the epithelial-to-mesenchymal transition (EMT)-associated gene expression, such as Vimentin and E-cadherin, in response to either oxidative stress or DNA damage [37]. Through JNK/p300 CBP signaling cascades, the trans-differentiation of the EMT pathway is a critical cellular event that controls the induction of cell apoptosis and migration in prostate, breast, and colon cancer cells; further studies are mediated actions in CIL-102-treated CRC cells.

Based on the proteomic differential displays of DLD-1 cells [21,22,27], our results showed the important finding that these activation effects result from a downstream gene of ERP29 and FUMH expression and the phosphorylation of the JNK/p300 CBP pathways, as well as the execution of apoptosis and anti-invasiveness by CIL-102 (Figure 5 and Figure 6 and Table 2). Interestingly, these results, from lenti shRNA ERP29 and shRNA FUMH alone CRC, showed ERP29 and FUMH protected CRC cells from a reduction of malignancy to promote metastasis and may be a potential effect of medical intervention for CRC therapy (Table 2). We recently researched the novel CIL-102-induced protein for either the induction of cell apoptosis or the inhibition of human DLD-1 cells and their molecular mechanism related to the targeted downstream protein; this was done by using two-dimensional electrophoresis (2-DE)-based proteomic analysis to identify the proteins involved in the activation of JNK/p300 CBP signaling pathway and oxidative stress. It suggested that the generation of ROS, as well as the JNK/p300 CBP pathway, to promote apoptosis and decrease invasiveness could be partly due to the Endoplasmic Reticulum resident Protein 29 (ERP29) and Fumarate hydratase (FUMH) expression by CIL-102. Thus, CIL-102, a derivative of Camptotheca acuminata, represents a novel chemotherapeutic agent worth investigating further. To validate these particular findings, further studies using other differential proteins are needed to determine whether there is mediated oxidative stress, DNA damage, and EMT pathway actions.

## 5. Conclusions

In conclusion, on the basis of proteomic differential proteins, we suggest the upregulation of ERP29 and FUMH expression by CIL-102 via the activation of the JNK/p300 CBP pathway and the induction of ROS production (Figure 7). This study is potentially interesting with respect to its novel chemotherapeutic approach of using CIL-102 to treat malignant CRC and cancer development.

## Figures and Tables

**Figure 1 biomolecules-11-01280-f001:**
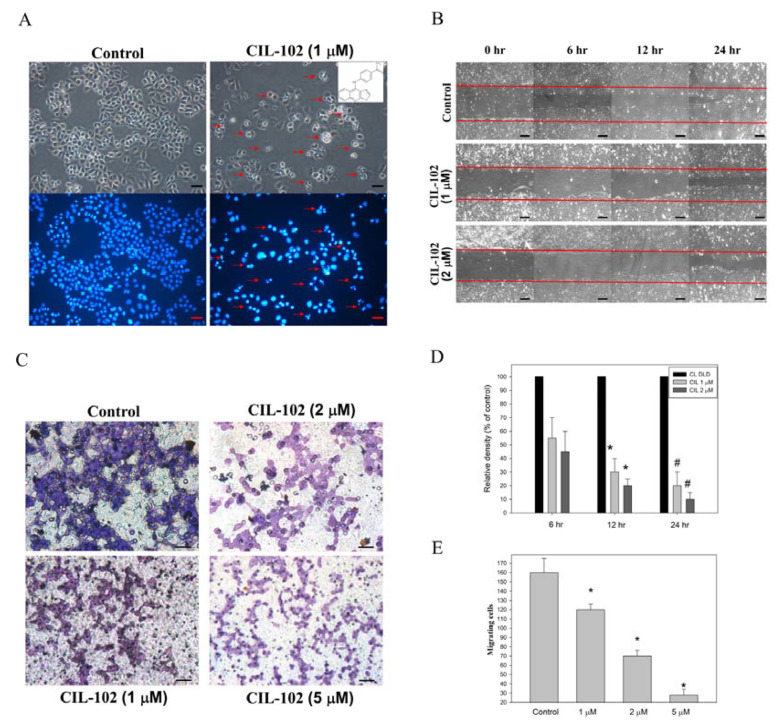
CIL-102 treatment impairs cell migration and the invasiveness of human colorectal cancer cells. (**A**) DLD-1 cells were treated with either 0.1% DMSO (as control) or CIL-102 (1 μM) for 24 h. The morphological characteristics (nuclear condensation) of the apoptotic cells were measured by DAPI staining under fluorescence microscopy. The red arrow indicates the apoptotic cells. (**B**,**D**) The migration of treated DLD-1 cells, with or without CIL-102 (1 and 2 μM), was performed for 6, 12, and 24 h and visualized by scratch-wound assay, as described in the Methods section. The quantification of the filled surface area with DLD-1 cells was done by using densitometric analyses. Data are presented as a percentage of the control group (means ± SD), based on three independent experiments in triplicate. Control VS CIL-102, * *p* < 0.05, compared with the control group for 12 h; ^#^ *p* < 0.01, compared with the control group for 24 h. (**C**,**E**) Invasiveness of DLD-1 cells treated with various concentrations (1, 2, 5 μM) of CIL-102 for 24 h was detected by the Boyden Chamber method, as described in the Methods section. The lower and upper chemotaxis cells were separated by a polycarbonate membrane. The representative images of cell invasion were detected through a layer of matrigel into the inner membrane under microscopy. Magnification× 200. The number of cell invasions into the inner membrane was quantified by manual counting and the data are presented as means ± SD. Control cells indicate the cells with a saline treatment. The experiments were performed in triplicate. * *p* < 0.05, compared to the untreated control cultures.

**Figure 2 biomolecules-11-01280-f002:**
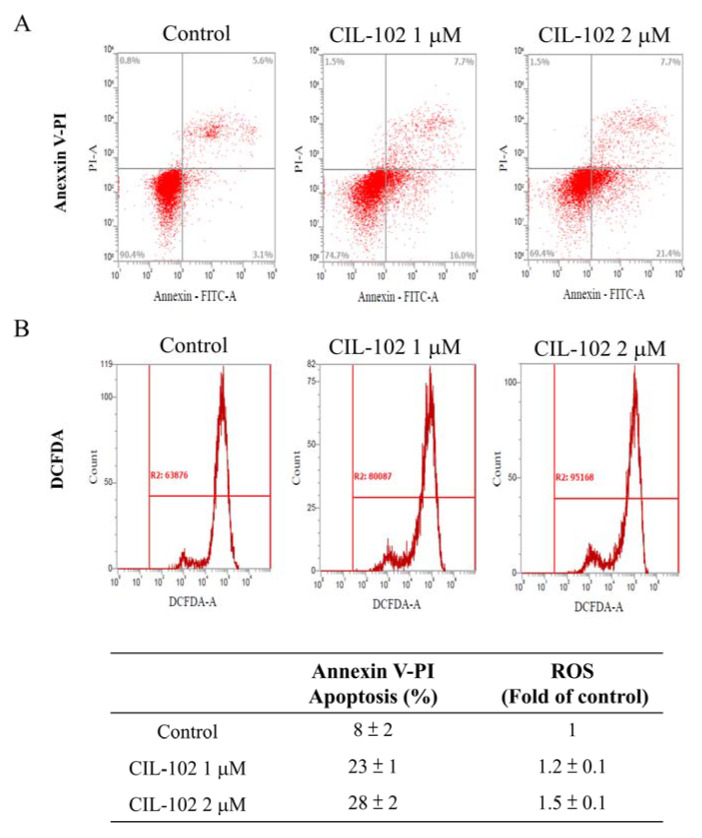
CIL-102 treatment induces cell apoptosis and the ROS production of DLD-1 cells. (**A**) FITC-conjugated Annexin-V and PI stainings of DLD-1 cells, with or without CIL-102 treatment, for 24 h were performed by flow cytometry analysis, as described in the Materials and Methods sections. The percentages of the apoptotic or neurosis cells in these treated cells are shown in each frame, as indicated. (**B**) Intracellular ROS of DLD-1 cells treated, with or without CIL-102, at 1- and 2-μM concentrations for 24 h was measured by a FACS analysis, as described in the Materials and Methods sections. Representative histograms of typical H2DCFDA profiles are shown. The production of ROS was expressed as the fold of the control group. Data are presented, based on three independent experiments, as mean ± S.D.

**Figure 3 biomolecules-11-01280-f003:**
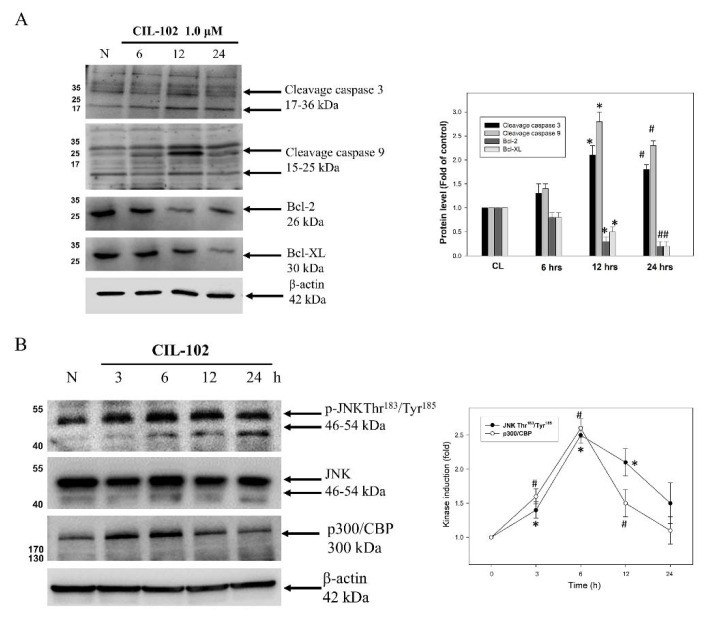
CIL-102 treatment activates the death-related signaling pathways in DLD-1 cells. The protein level of CIL-102 (1 µM)-treated DLD-1 cell for 0–24 h were detected by Western blot, including cleavage caspase 3, cleavage caspase 9, Bcl-2, and Bcl-XL (**A**) and p-JNK and p300/CBP (**B**). Protein levels were quantified by a densitometric analysis and normalized by a loading control, β-actin. The data are presented as 100% of the control group (means ± SD) from three independent experiments. * *p* < 0.05, when compared with the untreated control group at 12 h. ^#^
*p* < 0.05, when compared with the untreated control group at 24 h. (B). * *p* < 0.05, p-JNK when compared with the untreated control group at 12 h. ^#^
*p* < 0.05, p300/CBP when compared with the untreated control group at 24 h.

**Figure 4 biomolecules-11-01280-f004:**
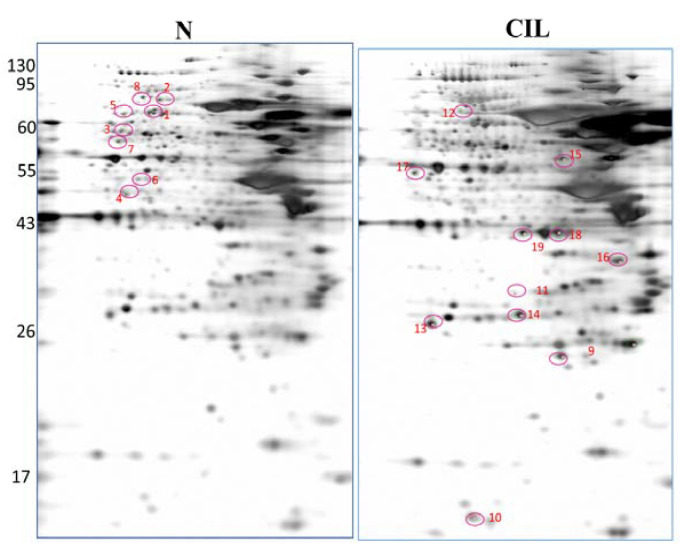
The representative two-dimensional gel electrophoretograms of DLD-1 cells, with or without the CIL-102 treatment. Six pairs of cell protein extracts from the CIL-102-treated and untreated DLD-1 cells were evaluated, and a representative pair of the proteomic gel images are shown. Nineteen protein spots (the upregulation of eight protein spots and the downregulation of 11 protein spots in the CIL-102-treated group) with a three-fold difference between both groups, were subjected to a MALDI-TOF-TOF analysis. The full names of these differentially displayed protein spots are encircled and annotated in Table 1, respectively.

**Figure 5 biomolecules-11-01280-f005:**
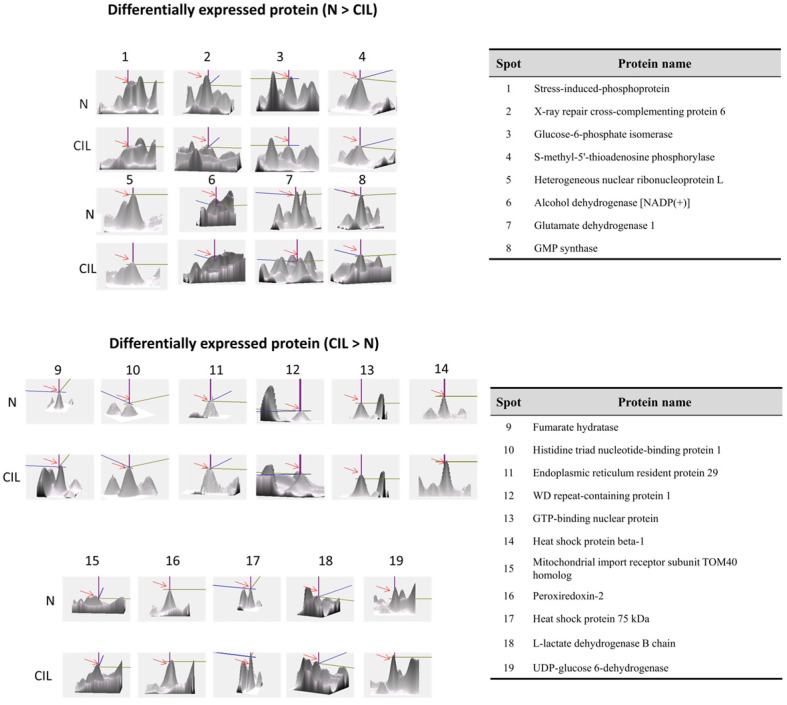
A representative close view of CIL-102-mediated differentially expressed proteins. The expression pattern of differentially expressed proteins (the number is the same as in Table 1) in DLD-1 cells, with or without CIL-102 treatment (one pair of experiments), are shown. Six pairs of cell protein extracts with four reproducible blots were performed in total.

**Figure 6 biomolecules-11-01280-f006:**
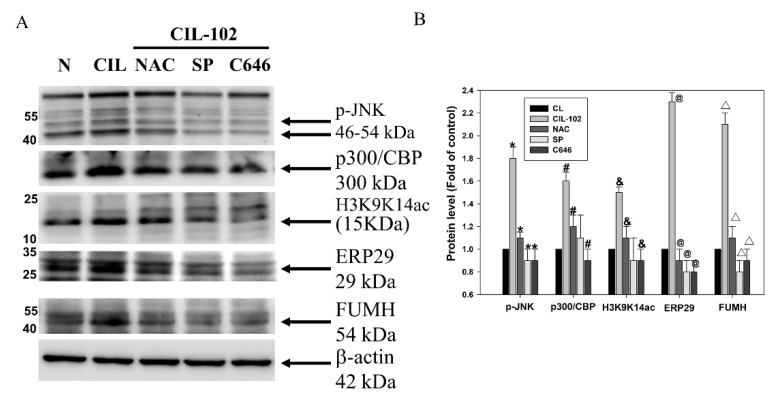
ROS production, histone acetylation, and JNK signal participate in the upregulation of ERP29 and FUMH by CIL-102. (**A**) DLD-1 cells were treated with kinase inhibitors (NAC for the inhibition of ROS production; SP is an inhibitor for the JNN signal; C646 is for the inhibition of acetyltransferase) in the presence or absence of CIL-102. One hour later, all cell lysates were prepared and subjected to a Western blot analysis. (**B**) The protein levels of phosphorylated JNK, p300/CBP and Histone H3 (H3K9K14ac), ERP29, FUMH, and β-actin were detected with the indicated antibodies and quantified by using a densitometric analysis. After normalization, β-actin served as loading control, and the data are presented as 100% of the untreated control group (means ± SD) from three independent experiments. * *p* < 0.05, p-JNK when compared with the untreated control group. # *p* < 0.05, p300/CBP when compared with the untreated control group. & *p* < 0.05, H3K9K14ac when compared with the untreated control group. @ *p* < 0.05, ERP29 when compared with the untreated control group. △ *p* < 0.05, FUMH when compared with the untreated control group.

**Figure 7 biomolecules-11-01280-f007:**
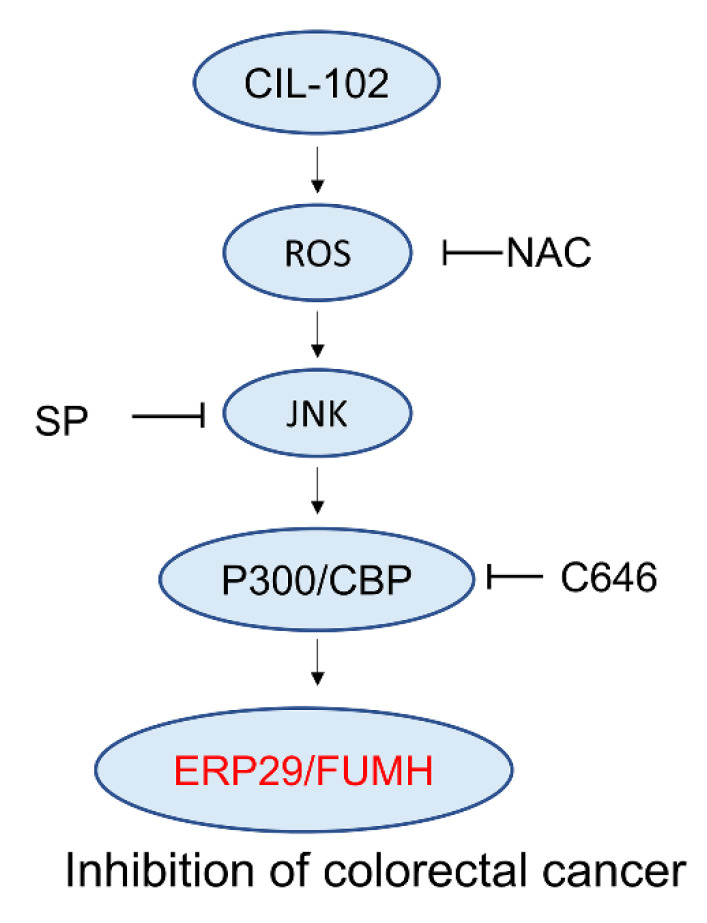
Working model of CIL-102 inhibits cell migration and the invasion of DLD-1 cells. ERP29 and FUMH are identified as novel CIL-102-upregulated proteins for the inhibition of cell migration and invasion in colorectal cancer. After the CIL-102 treatment, the activation of ROS and JNK/p300 CBP pathways are involved in cell migration and the invasion of colorectal cancer through the upregulation of ERP29 and FUMH.

**Table 1 biomolecules-11-01280-t001:** Differentially expressed protein.

Spot	Protein Name	Mr/PI	Accession No	MASCOT Score	Matched Peptides
1	Stress-induced-phosphoprotein	63/6.4	STIP1_HUMAN	1200	95
2	X-ray repair cross-complementing protein 6	70/6.2	XRCC6_HUMAN	1230	55
3	Glucose-6-phosphate isomerase	63/9.1	G6PI_HUMAN	60	2
4	S-methyl-5′-thioadenosine phosphorylase	31/6.9	MTAP_HUMAN	261	10
5	Heterogeneous nuclear ribonucleoprotein L	64/9.2	HNRPL_HUMAN	310	17
6	Alcohol dehydrogenase [NADP(+)]	36/6.3	AK1A1_HUMAN	438	17
7	Glutamate dehydrogenase 1	61/5.8	DHE3_HUMAN	1000	49
8	GMP synthase	77/6.4	GUAA_HUMAN	592	20
9	Fumarate hydratase	54/9.4	FUMH_HUMAN	420	21
10	Histidine triad nucleotide-binding protein 1	13/6.4	HINT1_HUMAN	125	8
11	Endoplasmic reticulum resident protein 29	29/7.5	ERP29_HUMAN	430	23
12	WD repeat-containing protein 1	66/6.1	WDR1_HUMAN	625	39
13	GTP-binding nuclear protein	24/7.7	RAN_HUMAN	534	40
14	Heat shock protein beta-1	22/5.9	HSPB1_HUMAN	411	42
15	Mitochondrial import receptor subunit TOM40 homolog	38/6.9	TOM40_HUMAN	372	15
16	Peroxiredoxin-2	22/5.5	PRDX2_HUMAN	548	38
17	Heat shock protein 75 kDa	80/8.9	TRAP1_HUMAN	468	14
18	L-lactate dehydrogenase B chain	36/5.6	LDHB_HUMAN	491	26
19	UDP-glucose 6-dehydrogenase	55/6.8	UGDH_HUMAN	1049	60

**Table 2 biomolecules-11-01280-t002:** Effects of the kinase inhibitor on the CIL-102 induction associated with cancer cell aggressive status in DLD-1 cells.

	Cell Invasion (%)	Migration (%)
Control	100	100
CIL-102	24 ± 2	42 ± 2
CIL-102 Lenti GFP	97 ± 2	98 ± 2
CIL-102 Lenti ERP29	55 ± 3	58 ± 3
CIL-102 Lenti FUMH	45 ± 3	65 ± 2
Lenti ERP29	165 ± 2	140 ± 2
Lenti FUMH	150 ± 2	145 ± 2

## Data Availability

All relevant data are within the paper.

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
