# Peer review of "Identification of Two Novel CIL-102 Upregulations of ERP29 and FUMH to Inhibit the Migration and Invasiveness of Colorectal Cancer Cells by Using the Proteomic Approach"

_biomolecules, 2021, doi:10.3390/biom11091280_

Round 1

Reviewer 1 Report

The paper entitled "Identification of two novel CIL-102-targeting proteins, ERP29 and FUMH, to inhibit the migration and invasiveness of colorectal cancer cells by using the proteomic approach is very interesting and the conclusion is supported by the obtained results.

Please read carefully the material and correct the misspelled words, even in the article title the word targeting is misspelled. Also, I would recommend to break figure 1 in 2 separate figures since the microscopy images are not visible at the current resolution.

Author Response

Author's Reply to the Review Report (Reviewer 1):
Reviewer #1:

Comments and Suggestions for Authors

The paper entitled "Identification of two novel CIL-102-targeting proteins, ERP29 and FUMH, to inhibit the migration and invasiveness of colorectal cancer cells by using the proteomic approach is very interesting and the conclusion is supported by the obtained results.

Please read carefully the material and correct the misspelled words, even in the article title the word targeting is misspelled. Also, I would recommend to break figure 1 in 2 separate figures since the microscopy images are not visible at the current resolution.

Response: We appreciate this comment and we have revised to the 600 dpi resolution version of the figure 1. In the manuscript as well as in the title, these are described, line 2 to line 4; fig 1 line 239 in the text.

Reviewer 2 Report

This manuscript reports the identification of a pathway through which one of the Campotheca acuminata alkaloid derivatives, CIL-102, exerts its anti-cancer effects on colorectal cancer cells, with the endoplasmic reticulum resident protein 29 (ERP29) and fumarate hydratase (FUMH) as key mediators. Two main issues should be addressed:

1- In several places in the manuscript as well as in the title, the authors write that ERP29 and FUMH are targets of CIL-102, which is incorrect. Drug targets are proteins directly, physically, affected by the drug through binding or modification. However, the authors present clear evidence that ERP29 and FUMH are much downstream, indirectly affected by CIL-102. Therefore, rewording is necessary to avoid confusion and draw conclusions that are properly supported by the data.

2-The authors should not discuss "data not shown", they need to show all their data in the manuscript. This includes data obtained with the HCoEpiC and HCT-116 cells, which support that CIL-102 functions on CRC cells, and not just one type, and does not affect normal cells. If the data are not included, then the authors should not mention them and should discuss the possibility that the observed effects are cell-dependent.

Author Response

RE: Revised version of biomolecules-1348756

Dear Ms. Florence Liu,

Enclosed please find one revised version entitled: Identification of two novel CIL-102-upregulation of ERP29 and FUMH to inhibit the migration and invasiveness of colorectal cancer cells by using the proteomic approach, which we would like to submit for publication in biomolecules.

This revised version has been carefully corrected according editor and referee’s reports point-by-point. We appreciate these valuable comments to strengthen our presentation. Please inform me if any revision is needed. The file marked change in blue color.

Furthermore, I would verify that no part of the manuscript is under consideration for publication elsewhere and it will not submit elsewhere if accepted by biomolecules and not before the Editorial Office has reached a decision.

Sincerely yours,

Ko-Chao Lee, MD.

Division of Colorectal Surgery,

Department of Surgery, Chang Gung Memorial Hospital, Kaohsiung Medical Center, Chang Gung University College of Medicine, Kaohsiung, Taiwan.

Telephone: +886-731-7123;

Fax: +886-731-8762

e-mail: [email protected];

Author's Reply to the Review Report (Reviewer 2):

Comments and Suggestions for Authors

This manuscript reports the identification of a pathway through which one of the Campotheca acuminata alkaloid derivatives, CIL-102, exerts its anti-cancer effects on colorectal cancer cells, with the endoplasmic reticulum resident protein 29 (ERP29) and fumarate hydratase (FUMH) as key mediators. Two main issues should be addressed:

1- In several places in the manuscript as well as in the title, the authors write that ERP29 and FUMH are targets of CIL-102, which is incorrect. Drug targets are proteins directly, physically, affected by the drug through binding or modification. However, the authors present clear evidence that ERP29 and FUMH are much downstream, indirectly affected by CIL-102. Therefore, rewording is necessary to avoid confusion and draw conclusions that are properly supported by the data.

Response: Thanks for the comments. In the manuscript as well as in the title, these are revised, line 2 to line 4; line 36, 85, 90, 220, 290, 304, 406, 413, 433, 434 and 454 in the text. The error in manuscript were revised in blue color.

2- The authors should not discuss "data not shown", they need to show all their data in the manuscript. This includes data obtained with the HCoEpiC and HCT-116 cells, which support that CIL-102 functions on CRC cells, and not just one type, and does not affect normal cells. If the data are not included, then the authors should not mention them and should discuss the possibility that the observed effects are cell-dependent.

Response:

We appreciate this comment and we have added a reference No. 13 to the revised version of the full text of manuscript from page 2 line 223.
